# Comparative Outcomes of Living and Deceased Donor Liver Transplantation in Adults: A Systematic Review and Meta-Analysis

**DOI:** 10.3390/jcm15010241

**Published:** 2025-12-28

**Authors:** Bestun Rashid, Mohammed Naga, Konrad Kobryń, Michał Grąt

**Affiliations:** Department of General, Transplant, and Liver Surgery, Medical University of Warsaw, 02-091 Warsaw, Poland; s087177@student.wum.edu.pl (B.R.); s087172@student.wum.edu.pl (M.N.); grat.michal@wum.edu.pl (M.G.)

**Keywords:** liver transplantation, living donor liver transplantation, deceased donor liver transplantation, postoperative complications, graft survival, patient survival, organ transplantation, organ donation

## Abstract

**Background/Objectives:** Living donor liver transplantation (LDLT) has emerged as an alternative to deceased donor liver transplantation (DDLT) in the circumstance of scarcity of deceased grafts. This systematic review and meta-analysis aims to compare outcomes between LDLT and DDLT in adult recipients. **Methods:** This systematic review was conducted in accordance with the PRISMA 2020 guidelines. A systematic literature search was performed using PubMed, EMBASE, and manual reference screening of relevant articles. We included peer-reviewed cohort studies comparing LDLT and DDLT in adult patients (≥18 years), published in English since 2015. **Results:** A total of 17 cohort studies, published between 2015 and 2024, fulfilled the inclusion criteria and were included in this systematic review and meta-analysis. These studies included a total of 22,514 adult liver transplant recipients, of whom 3832 (17.02%) and 18,682 (82.98%) underwent LDLT and DDLT, respectively. In comparison with DDLT, LDLT was associated with better 1-year patient survival, 5-year patient survival, and 5-year graft survival; however, these findings are based on low-certainty evidence and may be influenced by selection bias and baseline differences between cohorts. There were no significant differences between LDLT and DDLT groups in 3-year patient survival, 1-year graft survival, re-transplantation rates, biliary leakage, biliary stricture, or infection rates. **Conclusions:** LDLT is a valuable alternative to DDLT, particularly in regions with limited access to deceased donor organs, as it provides an excellent alternative to DDLT without compromising recipient outcomes, though further high-quality studies are needed.

## 1. Introduction

Liver transplantation (LT) is the treatment of choice for end-stage liver disease and selected liver malignancies. Since its introduction in 1963 [1], the demand for liver grafts has steadily increased, resulting in significant organ shortages, long waiting lists, and increased dropout risk [2]. To address the gap between supply and demand, alternative strategies such as grafts from extended criteria donors, donation after cardiac death, and living donor liver transplantation (LDLT) have been adopted. The development of LDLT following DDLT provided a crucial strategy to circumvent organ shortages [1,3]. Consequently, LDLT has become increasingly vital in regions with limited deceased donor availability and high waiting mortality.

The outcomes of LDLT and DDLT have been studied extensively, but findings remain inconsistent. Some meta-analyses report a survival advantage with LDLT [4], while other studies suggest comparable outcomes between the two techniques [5]. Moreover, comprehensive comparisons across both survival and post-transplant complications are limited, and newer data have not yet been fully synthesized. Furthermore, previous systematic reviews and meta-analyses comparing LDLT and DDLT have often focused on specific population subgroups, such as those with hepatocellular carcinoma or primary sclerosing cholangitis, or by addressing only a single outcome, such as patient survival [6,7]. This approach makes it difficult to establish a clear consensus of the relative outcomes for the general adult transplant population. To address this gap, this systematic review and meta-analysis aimed to comprehensively compare LDLT and DDLT in adult recipients (≥18 years) regardless of the underlying liver pathology warranting transplantation. We focused specifically on the following critical outcomes: overall patient survival, graft survival, re-transplantation rates, infection rates, and biliary complication rates.

## 2. Materials and Methods

This systematic review and meta-analysis was conducted and reported following the Preferred Reporting Items for Systematic Reviews and Meta-Analyses (PRISMA) 2020 guidelines. The research protocol was registered in the International Prospective Register of Systematic Reviews (PROSPERO), maintained by the Centre for Reviews and Dissemination (CRD), University of York, York, UK, (http://www.crd.york.ac.uk/PROSPERO) under registration number CRD420251044098. The protocol can be accessed at https://www.crd.york.ac.uk/PROSPERO/view/CRD420251044098 (last accessed on 11 May 2025). This study is a systematic review and meta-analysis comparing the outcomes of LDLT and DDLT in adult patients.

**Search strategy:** A systematic literature search was performed independently by two of the manuscript’s authors (Rashid and Naga) using PubMed and EMBASE. The final search was conducted on 28 March 2025 for PubMed and on 17 April 2025 for Embase. The search strategy was based on different combinations of words for each database. For PubMed, we used combinations of Medical Subject Headings (MeSH) and relevant free-text keywords. For Embase, equivalent concepts were searched using Emtree terms and free-text terms. In both databases, terms were grouped into thematic clusters addressing transplantation techniques, donor types, clinical outcomes, postoperative complications, graft rejection, and survival. These terms were then combined using Boolean operators (AND/OR) into three comprehensive search strategies, which were run independently in each database. The search strategy included filters to limit results to studies published in English within the past 10 years, involving adult human subjects. In addition to database searching, we screened the reference lists of relevant systematic reviews and meta-analyses to identify additional potentially eligible studies. All such studies were re-screened using our predefined inclusion and exclusion criteria. Conference proceedings and specific journals were not manually searched. Citation searching was not performed. No additional sources such as clinical trial registries, regulatory agency websites, pharmaceutical company data, or gray literature (e.g., theses, reports, or preprints) were searched. The full search strings used in both databases are presented in Appendix B.

**Study selection:** The same two authors independently screened the titles and abstracts of the studies that were identified from PubMed and Embase. Duplicate studies were excluded. We designed the following predefined eligibility criteria for the selection of studies. The following criteria were set for inclusion in this systematic review: (1) Adult patients (≥18 years) who underwent LDLT or DDLT, (2) Randomized controlled trials (RCTs) and cohort studies, (3) Studies that compare LDLT and DDLT outcomes, (4) Published in English, (5) Peer-reviewed articles, (6) Published within the last 10 years, and (7) Studies focused on adult-to-adult LDLT, either globally or region-specific (e.g., Asia, Europe, or North America). The following exclusion criteria were set:

(1) Pediatric liver transplant patients (<18 years), (2) Multi-organ transplant recipients, (3) Animal or in vitro studies, (4) Studies on auxiliary or split-liver transplantation, (5) Studies lacking relevant clinical outcomes (e.g., only focusing on economic factors without clinical data), (6) Non-English studies, (7) Gray literature (conference abstracts, unpublished data, theses, editorials, non-peer-reviewed literature), (8) Systematic reviews and meta-analyses, and (9) Studies published before 2015. No translation of non-English articles was required. Inter-reviewer agreement was quantified using Cohen’s kappa at both the title/abstract screening and full-text screening stages.

**Data extraction:** The same two authors independently extracted each included study. Discrepancies in the data extraction process were resolved through discussion. The following parameters were extracted: study characteristics (first author, region, year of publication, study design, follow-up period, and study period) and population characteristics (sample size, ages, gender, diagnoses of patients, and pre-transplant MELD score); moreover, specific postoperative complications, including biliary complications, hepatic artery thrombosis, infectious complications, 1-, 3-, and 5-year patient overall survival (OS), 1- and 5-year graft survival rate, and re-transplantation, were recorded and compared between LDLT and DDLT recipients. Follow-up durations were recorded and presented for descriptive purposes only; no adjustment or time-to-event standardization across studies was performed. Continuous variables were extracted for descriptive purposes only and are presented in their original format (mean ± SD or median [range]); no statistical conversion was performed. No duplicate or overlapping reports were identified for the included studies; therefore, no decision rules were required to resolve data inconsistencies across reports. No author was contacted for clarification or to obtain additional information. Missing or unclear data were recorded as not reported (NR). Prior to data extraction, the exclusion criterion regarding minimum sample size (<100 recipients) was removed to avoid unnecessary exclusion of valuable data from smaller studies and to enhance the comprehensiveness and representativeness of the included evidence.

**Quality assessment:** The Down and Black checklist (maximum score: 32) was used to assess the methodological quality of the included studies. Studies were categorized as high quality (22–32 points), moderate quality (16–21 points), or low quality (<16 points) based on their total scores. Two reviewers (Rashid B. and Naga M.) independently assessed each study across five domains: reporting, external validity, internal validity, and power. Discrepancies in scoring were resolved through discussion. To assess the risk of selective outcome reporting, we searched for registered protocols or pre-analysis plans for each included study across multiple sources, including Google Scholar, ISRCTN, and OSF Registries. No publicly available protocols were identified for any of the included studies, the majority of which were retrospective in design, and retrospective studies are not typically pre-registered.

**Statistical analysis:** The meta-analysis was conducted in accordance with the Cochrane Handbook for Systematic Reviews of Interventions, and statistical analyses were performed using Review Manager Web (RevMan Web; The Cochrane Collaboration). Dichotomous outcomes were analyzed by calculating odds ratios (ORs) with 95% confidence intervals (CIs) using the Inverse Variance method. A random-effects model was applied to all meta-analyses due to anticipated clinical and methodological diversity among included studies. Heterogeneity was quantified using the inconsistency statistic (I^2^). An I^2^ value of ≤25% was considered low heterogeneity, 26–50% moderate, and >50% substantial. The heterogeneity estimator used was the DerSimonian and Laird method. Summary effect confidence intervals were calculated using the Hartung–Knapp–Sidik–Jonkman (HKSJ) method to account for uncertainty in between-study variance estimates. For outcomes with fewer than 10 included studies, assessment of publication bias was not performed. For outcomes with 10 or more included studies, publication bias was assessed visually using funnel plots generated in JASP (version 0.19.1.0, University of Amsterdam), and statistically using Egger’s regression test performed in the same software. Sensitivity analyses were performed by sequentially excluding individual studies to assess the robustness of pooled estimates. No subgroup analysis or meta-regression was conducted to explore sources of heterogeneity due to the limited number of included studies and the lack of consistent study-level covariate data. The “Grading of Recommendations Assessment, Development and Evaluation” (GRADE) approach was used to assess the certainty of evidence for each outcome. A Summary of Findings table was generated using GRADEpro GDT to summarize effect estimates and certainty of evidence for the primary outcomes (Appendix A). Statistical significance was set at a *p*-value of <0.05.

## 3. Results

### 3.1. Study Selection

The literature review is summarized in a PRISMA diagram (Figure 1). The literature search yielded 1007 articles, of which 0 were duplicates. After removal of duplications, 1007 abstracts and titles were screened independently by two reviewers. From these, 26 articles were selected for full-text review based on the inclusion criteria. Additionally, 9 relevant articles were identified through manual reference searching, resulting in a total of 35 full-text articles assessed for eligibility. After independent review of the full texts, 18 articles were excluded because they were not relevant to the research question. Ultimately, 17 studies dated between 2015 and 2024 fulfilled the selection criteria and were therefore included in this systematic review; eight studies were from single centers, and nine included multi-center data. All studies were retrospective except for one prospective cohort study and one longitudinal study combining retrospective and prospective data; notably, only two of the retrospective studies employed propensity score matching. These studies were conducted in the east and in the west. No randomized controlled studies were identified. Characteristics of the included studies are summarized in Table 1 and Table 2. These articles included a total of 22,514 adult patients. Of these patients, 3832 (17.02%) and 18,682 (82.98%) underwent LDLT and DDLT, respectively. This corresponds to an approximate ratio of 1:5 between living donor and deceased donor liver transplantations. The Down and Black checklist was used to assess the methodological quality of the included studies, and the results are presented in Table 3. The reliability of the screening process was quantified as follows: inter-reviewer agreement during the title and abstract screening stage demonstrated substantial agreement, with a Cohen’s kappa of 0.729 (95% CI: 0.598–0.861) and an observed agreement of 98.51%. For the full-text eligibility screening, moderate agreement was observed, with a Cohen’s kappa of 0.599 (95% CI: 0.334–0.864) and an observed agreement of 80.00%. Discrepancies in selection were resolved through discussion.

**Table 1 jcm-15-00241-t001:** Characteristics of the Included Studies.

Study Name	Year	Country	Centers (Single or Multiple)	Study Design	Study Duration	Total Sample Size	LDLT Sample Size	DDLT Sample Size	Follow-Up Period
Ninomiya [8]	2015	Japan & USA (LDLT from Japan, DDLT from USA)	Multi-center	Retrospective Cohort	2002–2010	495	133	362	**Median** of 75 months (LDLT); 72 months (DDLT)
Al-Sebayel [9]	2015	KSA	Single	Retrospective Cohort	2001–2013	491	222	269	**Median** of 75 months (LDLT); 123 months (DDLT)
Hu [2]	2015	China	Multi-center	Retrospective Cohort	1999–2009	6860	389	6471	**Median** of 16.38 months (LDLT); 14.97 months (DDLT)
Samstein [10]	2016	USA	Multi-center	Prospective Observational	1998–2009	1036	565	471	**Median** of 57.6 months (LDLT); 50.28 months (DDLT)
Barbas [11]	2017	Canada	Single	Retrospective Cohort	2000–2014	176	48	128	**Median** of 52.9 months for both LDLT and DDLT
Chok [12]	2017	China	Single	Retrospective Cohort	2005–2014	94	54	40	**Median** of 77.4 months (LDLT); 55.4 months (DDLT)
Azoulay [13]	2017	France	Multi-center	Retrospective Cohort	2000–2009	699	79	620 (BDLT)	**Mean** of 70.4 ± 64.3 months; 102.5 ± 16 months (LDLT) and 67.2 ± 160 months (BDLT)]
E. Kim [14]	2017	South Korea	Single	Retrospective Cohort	2010–2014	185	109	76	12 years post-transplant or death for both LDLT and DDLT
J.M. Kim [15]	2017	Korea	Multi-center	Retrospective Cohort	1999–2012	62	31	31	**Median** of 36 months (LDLT); 18 months (DDLT)
Kwon [16]	2017	Korea	Single	Retrospective Cohort	2000–2017	25	15	10	N/A
Wong [17]	2019	China	Single	Retrospective Cohort	1995–2014	130	65	65	**Median** of 62 months for both LDLT and DDLT
Humar [18]	2019	USA	Single	Retrospective Cohort	2009–2019	837	245	592	**Median** of 33.5 months (LDLT); 53.8 months (DDLT)
Braun [19]	2020	USA	N/A (A2ALL cohort study)	Longitudinal Cohort (retrospective& prospective data)	1998–2014 (retrospective); 2004–2009 (prospective)	254	168	86	N/A
Ziogas [20]	2020	USA	Multi-center (OPTN database)	Retrospective Cohort	2002–2018	6911	912	5999	N/A
Goto [21]	2022	Canada	Single	Retrospective Cohort	2000–2019	2264	668	1596	**Median** of 56.4 years for both LDLT and DDLT
Amara [22]	2022	USA	Multi-center	Prospective Observational	2017–2019	1793	109	1684	N/A
Lapisatepun [23]	2024	Thailand	Multi-center	Retrospective Cohort	2014–2020	40	20	20	N/A

**Table 2 jcm-15-00241-t002:** Baseline Characteristics of Liver Transplant Recipients in the Included Studies.

Study Name	Year	Recipient Age (Years)		MELD Score		Recipient BMI		Recipient Male: Female		All Recipient Male %	Diagnosis
		LDLT	DDLT	LDLT	DDLT	LDLT	DDLT	LDLT	DDLT		
Ninomiya [8]	2015	**Mean** 57.6 ± 7.1	**Mean** 58.3 ± 7.4	**Mean** 11.9 ± 4.9	**Mean** 15.9 ± 8.3	**Mean** 23.9 ± 3.1	**Mean** 27.6 ± 5.4	78:55	285:77	73.3%	HCC
Al-Sebayel [9]	2015	**Median** 53 (15–80)	**Median** 52 (15–76)	18	16	N/A	N/A	139:83	153:116	59.5%	Mix
Hu [2]	2015	**Mean** 48.05 ± 8.65	**Mean** 50.09 ± 9.43	N/A	N/A	N/A	N/A	57:52	1135:549	66.3%	HCC
Samstein [10]	2016	**Mean** 51.0 ± 10.9	**Mean** 52.2 ± 10.4	**Mean** 15.2 ± 5.7	**Mean** 20.3 ± 8.9	**Mean** 26.5 ± 5.2	**Mean** 26.7 ± 4.9	311:254	285:186	57.52%	Mix
Barbers [11]	2017	**Mean** 54.7 ± 9.4	**Mean** 56.7 ± 9.3	**Mean** 17.8 ± 8.7	**Mean** 21.8 ± 10.3	**Mean** 29.7 ± 4.9	**Mean** 30.5 ± 6.4	35:13	87:41	69.3%	Mix
Chok [12]	2017	**Mean** 51± 12	**Mean** 51± 10.8	**Mean** 40± 1.3	**Mean** 39± 1.3	N/A	N/A	42:12	34:6	80.85%	Mix
Azoulay [13]	2017	**Mean** 54.3 ± 7.4	**Mean** 56.0 ± 7.8	**Mean** 14.7 ± 7.5	**Mean** 12.4 ± 6.5	N/A	N/A	65:14	677:106	86.1%	HCC
E. Kim [14]	2017	**Mean** 52.0 ± 8.5	**Mean** 53.1 ± 11.0	**Mean** 12.5 ± 8.3	**Mean** 24.9 ± 11.7	N/A	N/A	81:28	50:26	70.8%	Mix
J.M. Kim [15]	2017	N/A	N/A	**Median** 20 (11–40)	**Median** 20 (8–38)	N/A	N/A	21:10	21:10	67.7%	Mix
Kwon [16]	2017	**Mean** 71.7 ± 2.4	**Mean** 71.9 ± 1.9	**Mean** 14.1 ± 10.0	**Mean** 26.6 ± 8.9	N/A	N/A	11:4	8:2	76%	Mix
Wong [17]	2019	**Median** 55 (40–73)	**Median** 57 (41–67)	**Median** 11 (6–59)	**Median** 12 (6–37)	N/A	N/A	54:11	54:11	83.1%	HCC
Humar [18]	2019	**Mean** 56	**Mean** 56	**Mean** 16	**Mean** 22	**Mean** 28.4	**Mean** 29.7	145:100	414:178	67%	Mix
Braun [19]	2020	**Median** 53 (48–59)	N/A	**Median** 15 (13–19)	N/A	**Median** 26.2 (23.2–29.5)	N/A	119:49	N/A	70.8%	ALD
Ziogas [20]	2020	**Mean** 46.6 ± 13.9	Calculated mean 50.42 ± 13.19	**Mean** 15.2 ± 5.6	Calculated mean 23.08 ± 9.47	N/A	N/A	427:485	2857:3142	47.52%	CLD
Goto [21]	2022	**Median** 54 (47–61)	**Median** 57 (50–62)	**Median** 15 (12–20)	**Median** 16 (11–25)	**Median** 27 (23–29)	**Median** 28 (24–30)	402:266	1181:415	74% (DDLT); 60.2% (LDLT)	Mix
Amara [22]	2022	N/A	N/A	N/A	N/A	N/A	N/A	57:52	1135:549	66.3%	Mix
Lapisatepun [23]	2024	**Mean** 54.7 ± 11.7	**Mean** 48.8 ± 14.3	**Median** 14.5 (12–23.5)	**Median** 14.5 (7.5–22.5)	**Mean** 23.0 ± 2.6	**Mean** 24.3 ± 4.3	14:6	14:7	70%	Mix

**Table 3 jcm-15-00241-t003:** Methodological Quality Assessment of Included Studies Using the Downs and Black Checklist.

Study	Year	Total Score (out of 32)	Reporting	External Validity	Internal Validity Bias	Internal Validity Confounding	Power	Quality
Ninomiya [8]	2015	21	10/11	3/3	5/7	3/6	0/1	Moderate
Al-Sebayel [9]	2015	19	7/11	3/3	5/7	4/6	0/1	Moderate
Hu [2]	2015	21	9/11	3/3	5/7	4/6	0/1	Moderate
Samstein [10]	2016	22	10/11	3/3	5/7	4/6	0/1	High
Barbers [11]	2017	21	9/11	3/3	5/7	4/6	0/1	Moderate
Chok [12]	2017	22	10/11	3/3	5/7	4/6	0/1	High
Azoulay [13]	2017	22	9/11	3/3	5/7	5/6	0/1	High
E. Kim [14]	2017	21	9/11	3/3	5/7	4/6	0/1	Moderate
J.M. Kim [15]	2017	22	10/11	3/3	5/7	4/6	0/1	High
Kwon [16]	2017	22	10/11	3/3	5/7	4/6	0/1	High
Wong [17]	2019	22	10/11	3/3	5/7	4/6	0/1	High
Humar [18]	2019	20	10/11	3/3	4/7	3/6	0/1	Moderate
Braun [19]	2020	19	9/11	2/3	5/7	3/6	0/1	Moderate
Ziogas [20]	2020	19	8/11	3/3	5/7	3/6	0/1	Moderate
Goto [21]	2022	24	10/11	3/3	6/7	5/6	0/1	High
Amara [22]	2022	23	10/11	3/3	5/7	5/6	0/1	High
Lapesatepun [23]	2024	21	9/11	3/3	5/7	4/6	0/1	Moderate

**Figure 1 jcm-15-00241-f001:**
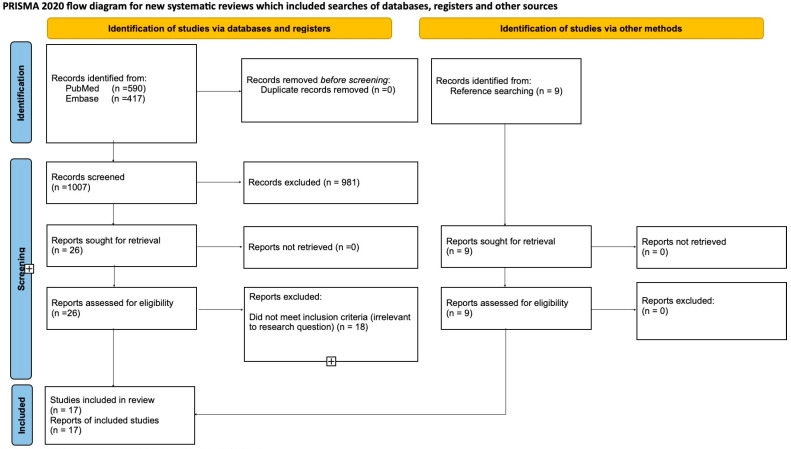
PRISMA 2020 flow diagram illustrating the study selection process. A total of 1007 records were identified through database searching (PubMed: *n* = 590; Embase: *n* = 417), with an additional 9 records identified through manual reference searching. After screening and eligibility assessment, 17 studies were included in the systematic review and meta-analysis [24].

### 3.2. One-Year Patient Survival

Eleven studies [2,8,11,12,14,15,16,17,20,21,23] comprising a total of 17,242 patients are included in the meta-analysis of one-year patient survival following LDLT versus DDLT. All of the included studies are retrospective cohort designs. According to the Downs and Black checklist, all studies contributing to this outcome are classified as having moderate quality, except for five studies rated as high quality (Table 3). The pooled odds ratio is 1.62 (95% *CI*: 1.08–2.43; *p* = 0.02), showing a statistically significant improvement in one-year survival for LDLT compared to DDLT (Figure 2). Significant heterogeneity is observed (*I*^2^ = 58%). Sensitivity analyses were conducted by sequentially excluding each study to assess the robustness of the findings. The pooled estimates ranged from 1.49 to 1.74 across the sensitivity analyses. Statistical significance was maintained in most analyses except when excluding Ninomiya (*OR* = 1.51, 95% *CI*: 0.98–2.34; *p* = 0.06), HU (*OR* = 1.49, 95% *CI*: 0.95–2.34; *p* = 0.08), or E. Kim (*OR* = 1.51, 95% *CI*: 0.99–2.30; *p* = 0.05), where significance was lost or borderline. Heterogeneity varied from 45% to 62% across analyses. These findings indicate that even though the overall effect favoring LDLT was generally robust, the result was sensitive to the inclusion of Ninomiya, HU, and E. Kim. Publication bias was assessed using a funnel plot, which appeared symmetrical, and Egger’s test did not suggest significant small-study effects (*p* = 0.221) (Figure 2). The certainty of evidence for this outcome was rated as very low, downgraded due to very serious inconsistency and serious indirectness and imprecision.

### 3.3. Three-Year Patient Survival

Seven studies [2,8,11,12,17,18,20] comprising a total of 15,503 patients are included in the meta-analysis of three-year patient survival following LDLT versus DDLT. All of the included studies are retrospective cohort designs. According to the Downs and Black checklist, all studies contributing to this outcome are classified as moderate quality, except for two studies rated as high quality (Table 3). The pooled odds ratio is 1.53 (95% CI: 0.96–2.44; *p* = 0.07), showing no statistically significant difference in three-year patient survival between the two groups (Figure 2). Significant heterogeneity is observed (I^2^ = 65%). Sensitivity analyses were conducted by sequentially excluding each study to evaluate the robustness of the findings. The pooled estimates ranged from 1.39 to 1.69 across the sensitivity analyses, with none reaching statistical significance except when excluding Wong (OR = 1.69, 95% CI: 1.28–2.24; *p* = 0.005). Heterogeneity varied from 43% to 71% across analyses. These findings indicate that even though the overall effect did not achieve statistical significance, the results were sensitive to the inclusion of individual studies, particularly Wong et al. As fewer than ten studies were included, formal assessment of publication bias was not performed. The certainty of evidence for this outcome was rated as very low, downgraded due to very serious inconsistency and serious indirectness and imprecision.

### 3.4. Five-Year Patient Survival

Ten studies [2,8,11,12,13,16,17,19,20,21] comprising a total of 17,908 patients are included in the meta-analysis of five-year patient survival following LDLT versus DDLT. All of the included studies are retrospective cohort designs, except for one longitudinal study that combined both retrospective and prospective elements. According to the Downs and Black checklist, half of the studies contributing to this outcome are classified as high quality, and the other half are rated as moderate quality (Table 3). The pooled odds ratio is 1.56 (95% *CI*: 1.16–2.10; *p* = 0.008), indicating a statistically significant survival benefit favoring LDLT over DDLT (Figure 2). Heterogeneity is significant (*I*^2^ = 64%). Sensitivity analyses were performed by sequentially excluding each study to evaluate the robustness of the findings. Across all sensitivity analyses, the pooled odds ratios ranged from 1.45 to 1.67, consistently demonstrating a significant survival advantage for LDLT, with all estimates maintaining statistical significance (*p* < 0.05). Heterogeneity varied from 43% to 68% across analyses. The direction and magnitude of the effect remained stable, confirming the robustness of the survival benefit associated with LDLT. Publication bias was assessed using a funnel plot, which appeared symmetrical, and Egger’s test did not suggest significant small-study effects (*p* = 0.303) (Figure 2). The certainty of evidence for this outcome was rated very low, downgraded due to very serious inconsistency and serious indirectness and imprecision.

### 3.5. One-Year Graft Survival

Five studies [11,12,16,20,21] comprising a total of 9470 patients are included in the meta-analysis of one-year graft survival following LDLT versus DDLT. All of the included studies are retrospective cohort designs. According to the Downs and Black checklist, all studies contributing to this outcome are classified as high quality, except for two studies, which are rated as moderate quality (Table 3). The pooled odds ratio is 0.96 (95% *CI*: 0.85–1.10; *p* = 0.47), indicating no statistically significant difference in one-year graft survival between the two groups (Figure 3). Heterogeneity is low (*I*^2^ = 0%), reflecting low variability among the included studies. Sensitivity analyses were performed by sequentially excluding each study to assess the robustness of the findings. The pooled estimates remained consistent across all sensitivity analyses, with odds ratios ranging from 0.91 to 1.03, and none achieving statistical significance. The direction and magnitude of effect were stable, and heterogeneity remained low (*I*^2^ = 0%) in all analyses. As the number of included studies was fewer than ten, formal assessment of publication bias using funnel plot or Egger’s test was not performed. The certainty of evidence for this outcome was rated as low, with no downgrades for risk of bias, inconsistency, indirectness, imprecision, or publication bias.

### 3.6. Five-Year Graft Survival

Six studies [11,12,16,19,20,21] comprising a total of 9724 patients are included in the meta-analysis of five-year graft survival following LDLT versus DDLT. All of the included studies are retrospective cohort designs, except for one longitudinal study that combined both retrospective and prospective elements. According to the Downs and Black checklist, half of the studies contributing to this outcome are classified as high quality, and the other half are rated as moderate quality (Table 3). The pooled odds ratio is 1.19 (95% CI: 1.14–1.24; *p* = 0.0001), indicating a statistically significant higher odds of 5-year graft survival in the LDLT group compared to DDLT (Figure 3). Heterogeneity is low (I^2^ = 0%), reflecting low variability among the included studies. Sensitivity analyses were performed by sequentially excluding each study to assess the robustness of the findings. The pooled estimates remained consistent across all sensitivity analyses, with odds ratios ranging from 1.17 to 1.20, all retaining statistical significance (*p* < 0.01 in all analyses). The direction and magnitude of effect were stable, and heterogeneity remained low (I^2^ = 0%) in all analyses. As the number of included studies was fewer than ten, formal assessment of publication bias using funnel plot or Egger’s test was not performed. The certainty of evidence for this outcome was rated as low, with no downgrades for risk of bias, inconsistency, indirectness, imprecision, or publication bias.

### 3.7. Re-Transplantation

Five studies [9,11,13,18,21], comprising 4467 patients, were included in the meta-analysis of re-transplantation rates comparing living donor liver transplantation (LDLT) and deceased donor liver transplantation (DDLT). All of the included studies were retrospective cohort designs. According to the Downs and Black checklist, all studies contributing to this outcome are classified as moderate quality, except for two studies, which are rated as high quality (Table 3). The pooled odds ratio was 1.46 (95% *CI*: 0.40–5.28; *p* = 0.46), indicating no statistically significant difference between the groups (Figure 4). Heterogeneity was substantial (*I*^2^ = 79%). Sensitivity analyses, performed by sequentially excluding individual studies, consistently yielded non-significant results, with pooled odds ratios ranging from 1.12 to 1.99 and wide confidence intervals overlapping the null. Heterogeneity remained moderate to substantial across sensitivity analyses (*I*^2^ = 45–84%). Moreover, due to the inclusion of fewer than ten studies, formal assessment of publication bias was not performed. The certainty of evidence for this outcome was rated as very low, downgraded due to very serious inconsistency and imprecision, and strongly suspected publication bias.

### 3.8. Hepatic Artery Thrombosis

Four studies [10,11,18,23] comprising 2089 patients were included in the meta-analysis of hepatic artery thrombosis. All studies included in this outcome were retrospective cohort designs, except for one prospective study. According to the Downs and Black checklist, all studies contributing to this outcome are classified as moderate quality, except for one study, which is rated as high quality (Table 3). The pooled odds ratio was 1.55 (95% *CI*: 0.97–2.47; *p* = 0.06), indicating no statistically significant difference in the odds of hepatic artery thrombosis between LDLT and DDLT (Figure 4). Heterogeneity was negligible (*I*^2^ = 0%), suggesting consistency among the included studies. Sensitivity analyses were conducted by sequentially excluding each study to evaluate the robustness of the findings. Exclusion of individual studies did not materially alter the direction or significance of the pooled effect; odds ratios ranged from 1.47 to 1.62, and statistical significance remained absent in all analyses except when Lapisatepun was excluded, which yielded a statistically significant association (*OR*: 1.62, 95% *CI*: 1.14–2.28; *p* = 0.03). Heterogeneity remained minimal (*I*^2^ = 0%) across all sensitivity analyses. Due to the inclusion of fewer than 10 studies, formal assessment of publication bias was not performed. The certainty of evidence for this outcome was rated as very low, downgraded due to very serious imprecision.

### 3.9. Biliary Stricture

Five studies [10,11,12,18,22] comprising 3936 patients were included in the meta-analysis of post-transplant biliary strictures. Three of the included studies were retrospective cohort designs, while the remaining two were prospective. According to the Downs and Black checklist, all studies contributing to this outcome are classified as high quality, except for two studies, which are rated as moderate quality (Table 3). The pooled odds ratio was 1.01 (95% *CI*: 0.40–2.56; *p* = 0.97), indicating no statistically significant difference in the odds of biliary strictures between LDLT and DDLT (Figure 4). Heterogeneity was substantial (*I*^2^ = 82%), reflecting considerable variability among the included studies. Sensitivity analyses were conducted by sequentially excluding each study to assess the robustness of the findings. Exclusion of individual studies did not materially alter the direction of the pooled effect; confidence intervals remained wide, and statistical significance was not achieved across all analyses. Heterogeneity decreased in some sensitivity analyses but remained moderate to high (*I*^2^ ranging from 9% to 86%). Due to the inclusion of fewer than 10 studies, formal assessment of publication bias was not performed. The certainty of evidence for this outcome was rated as very low, downgraded due to very serious inconsistency, and serious imprecision.

### 3.10. Biliary Leakage

Four studies [10,11,18,22] comprising 3842 patients were included in the meta-analysis of post-transplant biliary leakage. Two of the included studies were retrospective cohort designs, and the other two were prospective. According to the Downs and Black checklist, half of the studies contributing to this outcome are classified as high quality, and the other half are rated as moderate quality (Table 3). The pooled odds ratio was 6.19 (95% *CI*: 0.69–55.91; *p* = 0.08), indicating no statistically significant difference in the odds of biliary leakage between LDLT and DDLT (Figure 4). Heterogeneity was substantial (*I*^2^ = 96%), reflecting considerable variability among the included studies. Sensitivity analyses were conducted by sequentially excluding each study to assess the robustness of the findings. Exclusion of individual studies did not materially alter the direction of the pooled effect; however, confidence intervals widened, and statistical significance remained absent across all analyses. Heterogeneity remained high in all sensitivity analyses (*I*^2^ ranging from 65% to 98%). Due to the limited number of studies, formal assessment of publication bias was not performed. The certainty of evidence for this outcome was rated as very low, downgraded due to very serious inconsistency and imprecision, and strongly suspected publication bias.

### 3.11. Infections

Four studies [2,10,14,23] comprising 8121 patients were included in the meta-analysis of post-transplant infections. All of the included studies were retrospective cohort designs, except for one, which was prospective. According to the Downs and Black checklist, all studies contributing to this outcome are classified as moderate quality, except for one study, which is rated as high quality (Table 3). The pooled odds ratio was 0.53 (95% *CI*: 0.13–2.27; *p* = 0.26), indicating no statistically significant difference in the odds of infection between LDLT and DDLT (Figure 4). Heterogeneity was substantial (*I*^2^ = 90%), reflecting considerable variability across the included studies. Sensitivity analyses were conducted by sequentially excluding each study to assess the robustness of the findings. Exclusion of individual studies did not materially alter the direction of the pooled effect; however, confidence intervals widened, and statistical significance remained absent across all analyses. Heterogeneity remained high in all sensitivity analyses (*I*^2^ ranging from 82% to 93%). Due to the inclusion of fewer than 10 studies, formal assessment of publication bias was not performed. The certainty of evidence for this outcome was rated as very low, downgraded due to very serious inconsistency, serious imprecision, and strongly suspected publication bias.

## 4. Discussion

This systematic review and meta-analysis provides updated evidence on the comparative outcomes of living donor liver transplantation (LDLT) versus deceased donor liver transplantation (DDLT) in adult patients, synthesizing data from 17 studies involving over 22,000 recipients. The findings demonstrate that LDLT is associated with statistically significant improvements in one-year patient survival, five-year patient survival, and five-year graft survival. In contrast, no significant differences were observed between the two groups in terms of one-year graft survival, three-year patient survival, biliary complications (stricture and leakage), hepatic artery thrombosis, infection rates, or re-transplantation. However, the low certainty of evidence (GRADEpro) for the outcomes, along with the substantial heterogeneity, suggests that these results need to be interpreted with caution, as they are likely influenced by significant baseline differences between cohorts.

Our results partially align with previous meta-analyses such as those by Barbetta et al. [4] and Shingina et al. [25], which similarly reported superior or comparable survival outcomes for LDLT. Our analysis adds more recent data and includes additional outcomes, offering a broader synthesis of the available evidence.

While the overall clinical equivalence of LDLT and DDLT in terms of survival is encouraging, differences in donor selection, recipient acuity, and surgical expertise may partly explain the significant heterogeneity (I > 50%) in survival outcomes that were found in our study. In particular, the shorter cold ischemia time in LDLT is a crucial factor. Unlike DDLT grafts, which can undergo prolonged periods of cold storage, LDLT grafts are typically implanted within hours of procurement, therefore minimizing the ischemia–reperfusion injury associated with prolonged graft preservation, which is a well-known cause of early graft dysfunction and is negatively associated with post-transplant regeneration and survival [26,27]. The quality of the graft is superior in LDLT due to the rigorous selection process. Living donors are extensively screened healthy individuals, resulting in grafts that are steatosis-free and have functional hepatic mass [28]. This contrasts with DDLT grafts, which are obtained from extended criteria donors with comorbidities that impair the initial graft function and long-term outcomes [18,29]. Finally, the elective nature of LDLT allows for transplantation in recipients with lower MELD scores under optimized clinical conditions, avoiding the physiological decline that comes with long waiting times [4,18,30]. These factors combined result in the minimization of preservation injury, optimal graft function, and overall, a healthier recipient, explaining the survival benefit demonstrated in our meta-analysis.

The observed similarity in our study for complication rates—particularly for infections—aligns with previous findings. However, the results for biliary complications should be interpreted with caution, as the point estimate indicates a possible higher risk in LDLT despite statistical imprecision. This aligns with the established understanding that biliary complications remain a technical challenge in LDLT [4,31].

Although our analysis focuses on recipient outcomes, a crucial ethical consideration of LDLT is the risk assumed by the healthy living donor. It is essential to acknowledge that donor hepatectomy carries a non-negligible risk. A recent meta-analysis of over 60,000 donors revealed that the overall morbidity rate among living donors is 25%, with major complications occurring in 5.5% of cases [32]. Although rare, donor mortality does occur with an estimated rate of 0.06% [32]. These risks associated with LDLT procedure highlight the importance of rigorous donor selection, standardized surgical protocols, especially in high-volume centers, and comprehensive informed consent processes that clearly communicate these potential risks to ensure the donors’ safety and autonomy.

This systematic review and meta-analysis has several strengths. It adhered to PRISMA 2020 guidelines and was prospectively registered in PROSPERO. Inter-reviewer agreement was quantified using Cohen’s kappa at both the title/abstract screening and full-text screening stages. The Down and Black checklist was used to assess the methodological quality of the included studies, and the certainty of evidence was graded using the GRADE approach. Sensitivity analyses were conducted for all outcomes, and publication bias was assessed using funnel plots and Egger’s regression test for outcomes with ten or more studies.

However, the present study has several limitations, which should be considered when interpreting the results. By design, we required that eligible studies include a comparison cohort; as a consequence, studies from centers that exclusively performed either LDLT or DDLT were not included. All included studies were observational in design, with the majority being retrospective cohorts, which limited the power of the analysis; however, randomized controlled trials comparing LDLT and DDLT outcomes are impossible in clinical settings for obvious ethical reasons. In addition, different follow-up durations among the included studies were not adjusted for in the pooled analyses and may affect the comparability of outcome estimates. Included studies were conducted in different regions where policies and ethics about liver transplantation were different, and this might cause potential bias and limit the generalizability of our findings. Furthermore, the exclusion of non-English and gray literature may have introduced a potential selection bias and limited the comprehensiveness of the evidence base.

Additionally, there was significant heterogeneity among the studies, reflecting the differences in practice, protocols, and possibly in outcomes. No subgroup analysis or meta-regression was conducted to explore sources of heterogeneity due to the limited number of included studies and the lack of consistent study-level covariate data. Moreover, the definition of some complications was not clear or uniform in different studies. There were also inherent differences between LDLT and DDLT recipients in terms of age and MELD score. Finally, donor-related outcomes and center-level characteristics were not analyzed, and quality-of-life outcomes were rarely reported in the included studies. While publication bias was assessed for outcomes with ten or more studies, the risk of selective reporting bias could not be fully evaluated, as protocol or registry data were not available.

## 5. Conclusions

LDLT remains a valuable option for patients in need of liver transplantation, as it provides an excellent alternative to DDLT; the application of LDLT should be considered more, especially in areas with an extremely limited deceased donor pool. This review suggests that LDLT may be associated with similar outcomes as DDLT when it comes to re-transplantation rate, infection rate, biliary leakage, and biliary stricture. While our synthesis of available data suggests a potential survival advantage for LDLT, this is based on low-certainty evidence and is likely influenced by selection bias. LDLT is a valuable alternative with comparable or potentially better medium-term survival in selected patients, but the definitive claims of superiority require more robust, adjusted comparative studies.

## Figures and Tables

**Figure 2 jcm-15-00241-f002:**
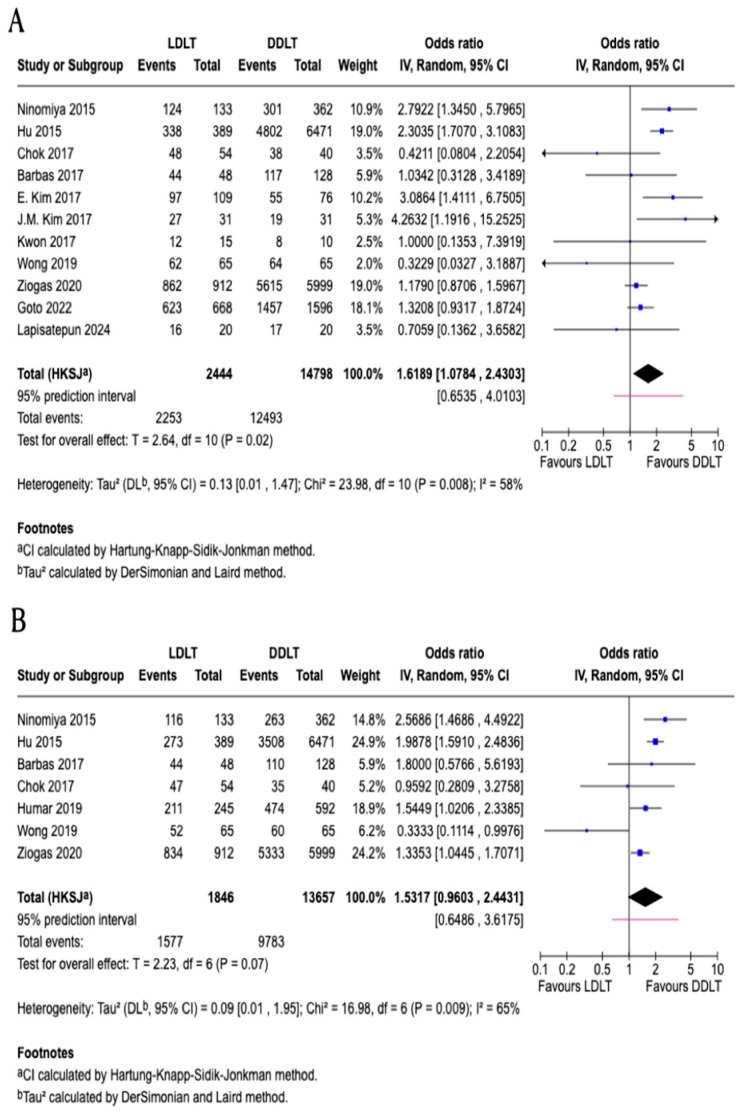
Forest plots and funnel plots comparing patient survival between living donor liver transplantation (LDLT) and deceased donor liver transplantation (DDLT). (**A**) One-year patient survival (11 studies [2,8,11,12,14,15,16,17,20,21,23]; *n* = 17,242; OR = 1.62, 95% CI: 1.08–2.43; *p* = 0.02; I^2^ = 58%). (**B**) Three-year patient survival (7 studies [2,8,11,12,17,18,20]; *n* = 15,503; OR = 1.53, 95% CI: 0.96–2.44; *p* = 0.07; I^2^ = 65%). (**C**) Five-year patient survival (10 studies [2,8,11,12,13,16,17,19,20,21]; *n* = 17,908; OR = 1.56, 95% CI: 1.16–2.10; *p* = 0.008; I^2^ = 64%). (**D**) Funnel plot for one-year patient survival (Egger’s test *p* = 0.221). (**E**) Funnel plot for five-year patient survival (Egger’s test *p* = 0.303). CI, confidence interval; OR, odds ratio.

**Figure 3 jcm-15-00241-f003:**
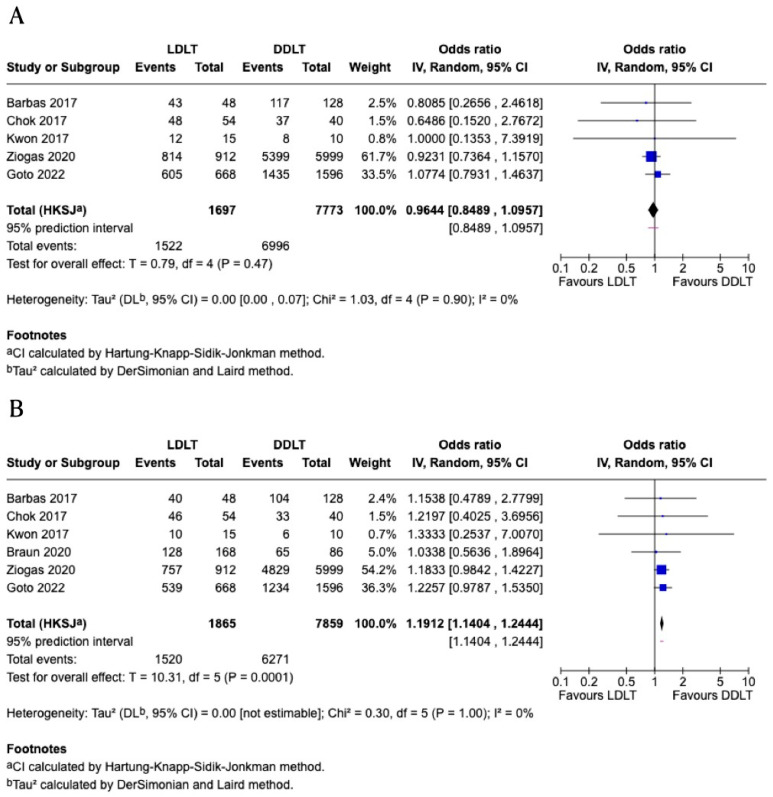
Forest plots comparing graft survival between living donor liver transplantation (LDLT) and deceased donor liver transplantation (DDLT). (**A**) One-year graft survival (5 studies [11,12,16,20,21]; *n* = 9470; OR = 0.96, 95% CI: 0.85–1.10; *p* = 0.47; I^2^ = 0%). (**B**) Five-year graft survival (6 studies [11,12,16,19,20,21]; *n* = 9724; OR = 1.19, 95% CI: 1.14–1.24; *p* = 0.0001; I^2^ = 0%). CI, confidence interval; OR, odds ratio.

**Figure 4 jcm-15-00241-f004:**
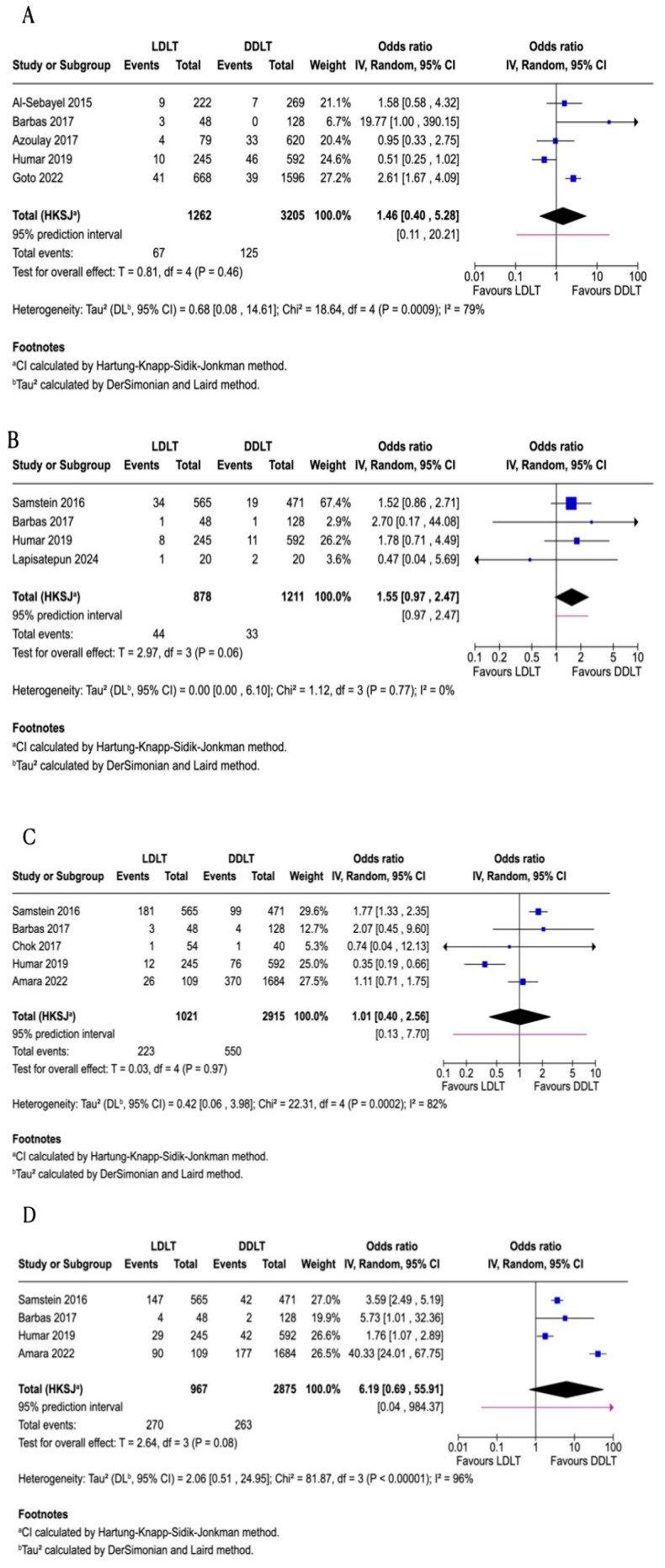
Forest plots comparing postoperative complications between living donor liver transplantation (LDLT) and deceased donor liver transplantation (DDLT). (**A**) Re-transplantation (5 studies [9,11,13,18,21]; *n* = 4467; OR = 1.46, 95% CI: 0.40–5.28; *p* = 0.46; I^2^ = 79%). (**B**) Hepatic artery thrombosis (4 studies [10,11,18,23]; *n* = 2089; OR = 1.55, 95% CI: 0.97–2.47; *p* = 0.06; I^2^ = 0%). (**C**) Biliary stricture (5 studies [10,11,12,18,22]; *n* = 3936; OR = 1.01, 95% CI: 0.40–2.56; *p* = 0.97; I^2^ = 82%). (**D**) Biliary leakage (4 studies [10,11,18,22]; *n* = 3842; OR = 6.19, 95% CI: 0.69–55.91; *p* = 0.08; I^2^ = 96%). (**E**) Infection (4 studies [2,10,14,23]; *n* = 8121; OR = 0.53, 95% CI: 0.13–2.27; *p* = 0.26; I^2^ = 90%). CI, confidence interval; OR, odds ratio.

## Data Availability

The original contributions presented in this study are included in the article/Appendix A. Further inquiries can be directed to the corresponding author.

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
