# Peer review of "Comparative Outcomes of Living and Deceased Donor Liver Transplantation in Adults: A Systematic Review and Meta-Analysis"

_jcm, 2025, doi:10.3390/jcm15010241_

Round 1
Reviewer 1 Report
Comments and Suggestions for Authors
The article by Rashid et al., entitled “Comparative Outcomes of Living and Deceased Donor Liver Transplantation in Adults: A Systematic Review and Meta-Analysis” is comprehensive review comparing LDLT (live donor liver transplantation) and DDLT (deceased donor liver transplantation) over graft survival. The review is of high significance as liver transplantation is the only option available for several critical liver diseases. Such systematic studies can help to improve the transplantation opportunities. However, the article can be improved by addressing the following suggestions.
- All Figures from 2 has some issue with the labeling and the figures. E.g., Figure 3 and 5 are missing with only labels available.
- Table 1 has variable follow-up periods for comparison. Not sure if this criteria is reported or addressed in the meta-analysis
- Overall, discussion on the living donor morbidity and mortality is missing in the article. A brief discussion acknowledging the donor risk or potential means to rectify can be included.
- Similarly, the ratio of total LDLT and DDLT transplantations is around 1:5. It can be mentioned in the article.
- It will be interesting to discuss why the 3-year survival data of LDLT after transplantation is different from the included, 1 & 5 years analyzed.
- The number of references at in a low end (only 22 references). Please address the reason for not including more than 22 references.
Reviewer 2 Report
Comments and Suggestions for Authors
Dear Authors,
Thank you for the opportunity to review your systematic review and meta-analysis. This is a commendable effort to synthesize a large volume of data on a critically important topic in transplantation. The manuscript is well-structured, follows PRISMA guidelines, and the statistical methodology is generally sound. The prospective registration and use of the GRADE framework are particular strengths.
However, in my view, several major points require revision to strengthen the manuscript and ensure the conclusions are robust and accurately reflect the data. My detailed comments are below.
Major Points:
-
Interpretation of Heterogeneity and Confounding in Survival Outcomes: the meta-analysis identifies substantial heterogeneity (I² > 50%) for the key survival outcomes. While this is noted statistically, the clinical implications are not sufficiently addressed in the discussion. This heterogeneity likely stems from significant baseline imbalances between LDLT and DDLT cohorts (e.g., consistently lower MELD scores in LDLT recipients, as shown in Table 2) and vast differences in center experience and geographic practice patterns. Action Required: The discussion must be revised to explicitly state that the observed survival benefits associated with LDLT are likely confounded by recipient selection bias (elective, lower-MELD transplants in LDLT vs. urgent, higher-MELD in DDLT). The conclusion of "superiority" should be tempered to reflect that the data show an "association" which may be driven by these factors. The very low GRADE certainty of evidence for these outcomes must be directly linked to the interpretation of the results.
-
Analysis and Discussion of Biliary Complications: the interpretation of the biliary leakage results is problematic. While the p-value is non-significant (p=0.08), the point estimate of an OR=6.19 indicates a >600% increased odds of biliary leak in the LDLT group. The wide confidence interval (0.69–55.91) indicates imprecision, not a null effect. Clinically, biliary leaks are a well-known technical challenge in LDLT. To state there is "no significant difference" is a misrepresentation of what the data suggests. Action Required: The discussion of biliary leakage must be rewritten. It should state that the results, while statistically inconclusive due to imprecision, point toward a clinically important and substantially higher risk of biliary leakage in LDLT, which is consistent with the established literature. Similarly, for biliary stricture (I² = 82%), the "non-significant" finding is uninformative due to extreme heterogeneity and this should be acknowledged as a limitation.
-
Alignment of Conclusions with Evidence: the strong conclusion that "LDLT is superior to DDLT in long term survivability" is not justified by the very low certainty of the evidence, as per your own GRADE assessment. The conclusions overreach the data, which are derived from heterogeneous, observational studies with significant confounding. Action Required: the abstract and main conclusions must be revised to be more nuanced. They should emphasize that while your synthesis of available data suggests a potential survival advantage for LDLT, this is based on low-certainty evidence and is likely influenced by selection bias. The primary conclusion should be that LDLT is a valuable alternative with comparable or potentially better medium-term survival in selected patients, but that definitive claims of superiority require more robust, adjusted comparative studies.
Minor Points:
-
Introduction: Can be condensed by shortening the historical details of the first DDLT and LDLT.
-
Methods: Please acknowledge the decision to exclude grey literature and non-English studies as a potential limitation.
-
Results (Figure 1): The report of 0 duplicates from 1007 articles is unusual. Please verify the de-duplication process.
-
Discussion: The statement that advancements have "mitigated" biliary risks is speculative and not directly supported by your data, which shows a strong signal for increased leaks. Please remove or rephrase.
Conclusion: your work addresses a vital question in transplantation. By addressing the points above—particularly by providing a more cautious and clinically nuanced interpretation of the survival and biliary complication data—you will significantly strengthen the manuscript and its contribution to the field.
I look forward to reviewing a revised version.
Comments on the Quality of English LanguageThe manuscript would benefit significantly from professional language editing. This process would enhance the flow, clarity, and overall professionalism of the writing without altering the scientific content. Many of the required corrections are minor, but their collective impact is substantial.
Reviewer 3 Report
Comments and Suggestions for Authors
This systematic review and meta-analysis addresses a highly relevant clinical issue: the comparative outcomes of Living Donor Liver Transplantation (LDLT) versus Deceased Donor Liver Transplantation (DDLT) in adults. The conclusion that LDLT is a valuable alternative, associated with better 1- and 5-year patient and 5-year graft survival, is of significant clinical interest. The review is conducted following PRISMA guidelines and synthesizes a large patient cohort (N=22,514). However, the methodological limitations of the primary literature, particularly the high risk of confounding bias, necessitate critical revisions to temper the certainty of the conclusions.
- The introduction effectively summarizes the background, but it would benefit from a clearer articulation of the study’s novelty and how it advances existing meta-analyses on LDLT vs. DDLT. Highlighting the unique contributions of this study would strengthen its significance.
- The inclusion of studies from 2015 onward is reasonable, but the rationale for this cutoff should be explained—was it chosen due to surgical technique evolution or data quality concerns? Similarly, removal of the minimum sample size threshold (<100) should be supported by justification to assure data representativeness.
- Many analyses report moderate to high heterogeneity (I² between 50–80%). The absence of subgroup or meta-regression analysis limits interpretability. It would be valuable to explore whether region (Asia vs. Western countries), study design (prospective vs. retrospective), or MELD score distribution accounts for observed variability.
- While survival outcomes are clearly presented, the discussion could more deeply analyze why LDLT may yield superior survival—such as reduced ischemia time, better graft quality, et al. Integrating mechanistic reasoning with evidence from the included studies would enhance the discussion’s depth.
- The numerous forest plots are informative but somewhat repetitive. Combining results into composite figures (e.g., patient survival vs. graft survival panels) and moving secondary or sensitivity analyses to supplementary materials could enhance clarity. Please ensure each figure caption includes the number of studies, total sample size, and p-values.
- The manuscript is generally well-written, but there are occasional stylistic redundancies and overly long sentences. A thorough language revision by a native English editor is recommended. Ensure consistent use of abbreviations (e.g., LDLT, DDLT, MELD), proper italicization of statistical symbols, and uniform referencing format.
- The limitations section could more explicitly address potential sources of bias such as retrospective design dominance, regional differences, and selection bias in donor and recipient characteristics. Discussing their possible effects on pooled results would add critical balance.
